# Magnetic Black Hole Thermodynamics in an Extended Phase Space with Nonlinear Electrodynamics

**DOI:** 10.3390/e26030261

**Published:** 2024-03-14

**Authors:** Sergey Il’ich Kruglov

**Affiliations:** 1Department of Physics, University of Toronto, 60 St. Georges St., Toronto, ON M5S 1A7, Canada; serguei.krouglov@utoronto.ca; 2Canadian Quantum Research Center, 204-3002 32 Ave., Vernon, BC V1T 2L7, Canada

**Keywords:** Einstein’s gravity, black holes, thermodynamics, Smarr relation, Gibbs free energy, Anti-de Sitter spacetime

## Abstract

We study Einstein’s gravity coupled to nonlinear electrodynamics with two parameters in anti-de Sitter spacetime. Magnetically charged black holes in an extended phase space are investigated. We obtain the mass and metric functions and the asymptotic and corrections to the Reissner–Nordström metric function when the cosmological constant vanishes. The first law of black hole thermodynamics in an extended phase space is formulated and the magnetic potential and the thermodynamic conjugate to the coupling are obtained. We prove the generalized Smarr relation. The heat capacity and the Gibbs free energy are computed and the phase transitions are studied. It is shown that the electric fields of charged objects at the origin and the electrostatic self-energy are finite within the nonlinear electrodynamics proposed.

## 1. Introduction

It is understood that the black hole area plays the role of entropy and the temperature is connected with the surface gravity [1,2,3,4,5]. The importance of gravity in AdS spacetime is due to the holographic principle (a gauge duality description) [6], which has applications in condensed matter physics. Firstly, black hole phase transitions in Schwarzschild–AdS spacetime were studied in Ref. [7]. The negative cosmological constant, in an extended-phase-space black hole thermodynamics, is linked with the thermodynamic pressure conjugated to the volume [8,9,10,11]. The cosmological constant variation was included in the first law of black hole thermodynamics in Refs. [12,13,14,15,16,17]. However, within general relativity, the cosmological constant Λ is a fixed external parameter. Moreover, such variation in Λ in the first law of black hole thermodynamics means the consideration of black hole ensembles possessing different asymptotics. This point of view is different from that of standard black hole thermodynamics, where the parameters are varied but the AdS background is fixed. There are some reasons to consider the variation in Λ in black hole thermodynamics. First of all, physical constants can arise as the vacuum expectation values are not fixed a priori and therefore may vary. As a result, these ‘constants’ are not real constants and may be included in the first law of black hole thermodynamics [18,19]. The second reason is that, without varying the cosmological constant, the Smarr relation is inconsistent with the first law of black hole thermodynamics [13]. When Λ is inserted in the first law of black hole thermodynamics, the mass M of the black hole should be treated as enthalpy but not internal energy [13]. The first law of black hole thermodynamics can be formulated within Einstein’s gravity if one includes the VdP term. This requires us to introduce a negative cosmological constant Λ as a positive pressure P=−Λ/(8π). As a result, we come to AdS spacetime. It should be noted that the thermodynamic pressure *P* is different from the local pressure that is present in the energy–momentum tensor. The conjugate variable to *P* is the thermodynamic volume $V=4\pi r_{+}^{3}/3$, where $r_{+}$ is the event horizon radius of a black hole.

In this paper, we study Einstein–AdS gravity coupled to nonlinear electrodynamics (NED) with two parameters, proposed here, that allow us to smooth out singularities. The first NED was Born–Infeld electrodynamics [20]; without the singularities of point-like particles possessing finite electric self-energy at the weak-field limit, it is converted into Maxwell’s theory. Our NED model has similar behavior. We study magnetic black holes and their thermodynamics in Einstein–AdS gravity in the extended phase space. The NED model, with coupling β and dimensionless parameter σ, proposed here, is of interest because it includes model [21] for σ=1. This unified approach allows us to find similarities and differences for different models.

The structure of the paper is as follows. In Section 2, we find the mass and metric functions and their asymptotics. Corrections to the Reissner–Nordström metric function are obtained when the cosmological constant is zero. We prove the first law of black hole thermodynamics in an extended phase space and obtain the magnetic potential and the thermodynamic conjugate to the coupling. The generalized Smarr formula is proven. The Gibbs free energy is calculated and depicted for some parameters and the phase transitions are studied in Section 3. In Appendix A, we obtain the electric fields of charged objects and corrections to Coulomb’s law. We show that the electrostatic self-energy of charged particles is finite in Appendix B. In Appendix C, we obtain the metric that is a solution to the Einstein–Maxwell system. Section 4 is a discussion of the results obtained.

We use units with $c=\hbar =k_{B}=1$.

## 2. Einstein–AdS Black Hole Solution

The action of Einstein’s gravity in AdS spacetime is given by
(1)$$I=\int d^{4}x\sqrt{-g}\left(\frac{R-2\Lambda }{16\pi G}+L\left(F\right)\right),$$
where *G* is the gravitational constant, $\Lambda =-3/l^{2}$ is the negative cosmological constant, and *l* is the AdS radius. We propose the NED Lagrangian as follows:(2)$$L\left(F\right)=-\frac{F}{4\pi {\left(1+2\beta F\right)}^{\sigma }},$$
where $F=F^{\mu \nu }F_{\mu \nu }/4=\left(B^{2}-E^{2}\right)/2$ is the Lorenz invariant and *E* and *B* are the electric and magnetic fields, correspondingly. The coupling β>0 has the dimension $L^{4}$, and the dimensionless parameter σ>0. The weak-field limit of Lagrangian (2) is Maxwell’s Lagrangian. Lagrangian (2) at σ=1 becomes the rational NED Lagrangian [22]. The NED Lagrangian (2) for some values of σ was used in the inflation scenario [23,24,25]. From action (1), one finds the Einstein and field equations
(3)$$R_{\mu \nu }-\frac{1}{2}g_{\mu \nu }R+\Lambda g_{\mu \nu }=8\pi GT_{\mu \nu },$$
(4)$$\partial_{\mu }\left(\sqrt{-g}L_{F}F^{\mu \nu }\right)=0,$$
where $L_{F}=\partial L\left(F\right)/\partial F$. The energy–stress tensor reads
(5)$$T_{\mu \nu }=F_{\mu \rho }F_{\nu }^{\;\rho }L_{F}+g_{\mu \nu }L\left(F\right).$$
We consider here spherical symmetry with the line element
(6)$$ds^{2}=-f\left(r\right)dt^{2}+\frac{1}{f(r)}dr^{2}+r^{2}\left(d\theta^{2}+\sin^{2}\left(\theta \right)d\phi^{2}\right).$$
Magnetic black holes possess the magnetic field $B=q/r^{2}$, where *q* is the magnetic charge. The metric function is given by (see Appendix C and [26])
(7)$$f\left(r\right)=1-\frac{2m(r)G}{r},$$
with the mass function
(8)$$m\left(r\right)=m_{0}+4\pi \int \rho \left(r\right)r^{2}dr,$$
where $m_{0}$ is an integration constant (the Schwarzschild mass) and ρ is the energy density. We obtain the energy density
(9)$$\rho =\rho_{M}-\frac{3}{8\pi Gl^{2}},$$
where the magnetic energy density found from Equations (2) and (5) is
$$\rho_{M}=\frac{q^{2}r^{4(\sigma -1)}}{8\pi {\left(r^{4}+q^{2}\beta \right)}^{\sigma }}.$$
Making use of Equations (8) and (9), we obtain the mass function
(10)$$m\left(r\right)=m_{0}+\frac{q^{2}r^{4\sigma -1}}{2\left(4\sigma -1\right){\left(q^{2}\beta \right)}^{\sigma }}F\left(\sigma -\frac{1}{4},\sigma ;\sigma +\frac{3}{4};-\frac{r^{4}}{q^{2}\beta }\right)-\frac{r^{3}}{2Gl^{2}},$$
where F(a,b;c;z) is the hypergeometric function. The magnetic energy is given by
(11)$$m_{M}=4\pi \int_{0}^{\infty }\rho_{M}\left(r\right)r^{2}dr=\frac{q^{3/2}\Gamma \left(\sigma -1/4\right)\Gamma \left(5/4\right)}{2\beta^{1/4}\Gamma \left(\sigma \right)},$$
where Γ(x) is the Gamma function. Equation (Equation 11) shows that, at Maxwell’s limit β→0, the black hole’s magnetic mass diverges. Therefore, a smooth limit to Maxwell’s theory is questionable. From Equations (7) and (10), one finds the metric function
(12)$$f\left(r\right)=1-\frac{2m_{0}G_{N}}{r}-\frac{q^{2}Gr^{4\sigma -2}}{\left(4\sigma -1\right){\left(q^{2}\beta \right)}^{\sigma }}F\left(\sigma -\frac{1}{4},\sigma ;\sigma +\frac{3}{4};-\frac{r^{4}}{q^{2}\beta }\right)+\frac{r^{2}}{l^{2}}.$$
We employ the relation [27]
(13)$$F\left(a,b;c;z\right)=1+\frac{ab}{c}z+\frac{a(a+1)b(b+1)}{c(c+1)}z^{2}+\ldots .,$$
for |z|<1, which will be used to obtain the asymptotic of the metric function as r→0. When the Schwarzschild mass is zero ($m_{0}=0$) and as r→0, the asymptotic is
(14)$$f\left(r\right)=1+\frac{r^{2}}{l^{2}}-\frac{Gq^{2}r^{4\sigma -2}}{{\left(q^{2}\beta \right)}^{\sigma }\left(4\sigma -1\right)}+\frac{G\sigma r^{4\sigma +2}}{\beta^{\sigma +1}q^{2\sigma }\left(4\sigma +3\right)}+O\left(r^{4\sigma +6}\right).$$
Equation (Equation 14) shows that, at σ≥1/2, a singularity of the metric function f(r) is absent. In addition, to avoid a conical singularity at r=0, we also should set 4σ−2>1 (σ>3/4). It is worth noting that the magnetic energy density $\rho_{M}$ is finite at r=0 only if σ≥1. Therefore, to have regular black holes, one has to assume that σ≥1. Then, from Equation (Equation 14), we find f(0)=1, which is a necessary condition to have regular spacetime. We explore the transformation [27]
$$F\left(a,b;c;z\right)=\frac{\Gamma (c)\Gamma (b-a)}{\Gamma (b)\Gamma (c-a)}{\left(-z\right)}^{-a}F\left(a,1-c+a;1-b+a;\frac{1}{z}\right)$$
(15)$$+\frac{\Gamma (c)\Gamma (a-b)}{\Gamma (a)\Gamma (c-b)}{\left(-z\right)}^{-b}F\left(b,1-c+b;1-a+b;\frac{1}{z}\right),$$
to obtain the asymptotic of the metric function as r→∞. By virtue of Equations (13) and (15), we find
(16)$$f\left(r\right)=1-\frac{2(m_{0}+m_{M})G}{r}+\frac{q^{2}G}{r^{2}}F\left(\sigma ,\frac{1}{4};\frac{5}{4};-\frac{q^{2}\beta }{r^{4}}\right)+\frac{r^{2}}{l^{2}},$$
where the relations Γ(1+z)=zΓ(z) and $F\left(a,0;c;z\right)=1$ are used. Making use of Equations (13) and (16) as r→∞ when the cosmological constant vanishes (l→∞), we find
(17)$$f\left(r\right)=1-\frac{2MG}{r}+\frac{q^{2}G}{r^{2}}-\frac{q^{4}\beta \sigma G}{5r^{6}}+O\left(r^{-10}\right),$$
where $M=m_{0}+m_{M}$ is the ADM mass (the total black hole mass as r→∞). It follows from Equation (Equation 17) that the corrections to the Reissner–Nordström solution are in the order of $O(r^{-6})$. When β→0, metric function (17) is converted into the Reissner–Nordström metric function. The plot of metric function (12) is given in Figure 1 with $m_{0}=0$, G=q=1, β=0.1, l=5.

In accordance with Figure 1, if parameter σ increases, the event horizon radius $r_{+}$ decreases. Figure 1 shows that black holes can have one or two horizons. It should be noted that when we set G=c=ℏ=1 as in Figure 1, we come to Planckian units [28]. Then, in this case, if one has, for example, dimensionless event horizon radius $r_{+}=1$ (as in Figure 1), in the usual units, $r_{+}=l_{Pl}={\left(G\hbar /c^{3}\right)}^{1/2}=1.616\times 10^{-33}$ cm, where $l_{Pl}$ is Planck’s length. When the dimensionless mass is m=2, for example, in the usual units, $m=2\times m_{Pl}=2\times {\left(\hbar c/G\right)}^{1/2}=2\times 2.177\times 10^{-5}$ g, where $m_{Pl}$ is Planck’s mass. Because, in Figure 1 the event horizon radius is small, we have here the example of tiny black holes (primordial black holes). Such black holes could have been created after the Big Bang. It is worth noting that such an example of quantum-sized black holes is described here by semiclassical gravity.

## 3. First Law of Black Hole Thermodynamics

The pressure, in extended-phase-space thermodynamics, is defined as P=−Λ/(8π) [13,14,17,29]. The coupling β is treated as the thermodynamic value and the mass *M* is a chemical enthalpy so that M=U+PV and *U* is the internal energy. In the following, we will use Planckian units with G=c=ℏ=1. By using Euler’s dimensional analysis [13,30], we have dimensions [M]=L, $\left[S\right]=L^{2}$, $\left[P\right]=L^{-2}$, $\left[J\right]=L^{2}$, [q]=L, $\left[\beta \right]=L^{2}$ and
(18)$$M=2S\frac{\partial M}{\partial S}-2P\frac{\partial M}{\partial P}+2J\frac{\partial M}{\partial J}+q\frac{\partial M}{\partial q}+2\beta \frac{\partial M}{\partial \beta },$$
where *J* is the black hole’s angular momentum. In the following, we consider non-rotating black holes and, therefore, J=0. The thermodynamic conjugate to coupling β is B=∂M/∂β (so-called vacuum polarization) [10]. The black hole volume *V* and entropy *S* are defined as
(19)$$V=\frac{4}{3}\pi r_{+}^{3},\;\;\;\;\;\;S=\pi r_{+}^{2}.$$
From Equation (Equation 16) and the equation $f(r_{+})=0$, where $r_{+}$ is the event horizon radius, one finds
(20)$$M\left(r_{+}\right)=\frac{r_{+}}{2}+\frac{q^{2}}{2r_{+}}F\left(\sigma ,\frac{1}{4};\frac{5}{4};-\frac{q^{2}\beta }{r_{+}^{4}}\right)+\frac{r_{+}^{3}}{2l^{2}}.$$
Making use of Equation (Equation 20), we obtain
$$dM\left(r_{+}\right)=[\frac{1}{2}+\frac{3r_{+}^{2}}{2l^{2}}-\frac{q^{2}}{2r_{+}^{2}}F\left(\sigma ,\frac{1}{4};\frac{5}{4};-\frac{q^{2}\beta }{r_{+}^{4}}\right)$$
$$+\frac{2\sigma q^{4}\beta }{5r_{+}^{6}}F\left(\sigma +1,\frac{5}{4};\frac{9}{4};-\frac{q^{2}\beta }{r_{+}^{4}}\right)]dr_{+}-\frac{r_{+}^{3}}{l^{3}}dl$$
$$+\left[\frac{q}{r_{+}}F\left(\sigma ,\frac{1}{4};\frac{5}{4};-\frac{q^{2}\beta }{r_{+}^{4}}\right)-\frac{q^{3}\beta \sigma }{5r_{+}^{5}}F\left(\sigma +1,\frac{5}{4};\frac{9}{4};-\frac{q^{2}\beta }{r_{+}^{4}}\right)\right]dq$$
(21)$$-\frac{q^{4}\sigma }{10r_{+}^{5}}F\left(\sigma +1,\frac{5}{4};\frac{9}{4};-\frac{q^{2}\beta }{r_{+}^{4}}\right)d\beta .$$
Here, we have used the relation [27]
(22)$$\frac{dF(a,b;c;z)}{dz}=\frac{ab}{c}F\left(a+1,b+1;c+1;z\right).$$
Defining the Hawking temperature
(23)$$T=\frac{f^{\prime }{\left(r\right)|}_{r=r_{+}}}{4\pi },$$
where $f^{\prime }\left(r\right)=\partial f\left(r\right)/\partial r$, and by virtue of Equations (16) and (23), we obtain
(24)$$T=\frac{1}{4\pi }\left[\frac{1}{r_{+}}+\frac{3r_{+}}{l^{2}}-\frac{q^{2}}{r_{+}^{3}}F\left(\sigma ,\frac{1}{4};\frac{5}{4};-\frac{q^{2}\beta }{r_{+}^{4}}\right)+\frac{4\sigma q^{4}\beta }{5r_{+}^{7}}F\left(\sigma +1,\frac{5}{4};\frac{9}{4};-\frac{q^{2}\beta }{r_{+}^{4}}\right)\right].$$
At β=0 in Equation (Equation 24), one finds the Maxwell–AdS black hole Hawking temperature. The first law of black hole thermodynamics follows from Equations (19), (20) and (24),
(25)dM=TdS+VdP+Φdq+Bdβ.
From Equations (21) and (25), we obtain the magnetic potential Φ and the thermodynamic conjugate to coupling β (vacuum polarization) B
$$\Phi =\frac{q}{r_{+}}F\left(\sigma ,\frac{1}{4};\frac{5}{4};-\frac{q^{2}\beta }{r_{+}^{4}}\right)-\frac{q^{3}\beta \sigma }{5r_{+}^{5}}F\left(\sigma +1,\frac{5}{4};\frac{9}{4};-\frac{q^{2}\beta }{r_{+}^{4}}\right),$$
(26)$$B=-\frac{q^{4}\sigma }{10r_{+}^{5}}F\left(\sigma +1,\frac{5}{4};\frac{9}{4};-\frac{q^{2}\beta }{r_{+}^{4}}\right).$$
The plots of Φ and B versus $r_{+}$ are depicted in Figure 2.

Figure 2, in the left panel, shows that as $r_{+}\to \infty$, the magnetic potential vanishes (Φ(∞)=0), and at $r_{+}=0$, it is finite. If the parameter σ increases, Φ(0) decreases. According to the right panel of Figure 2, at $r_{+}=0$, the vacuum polarization is finite, and as $r_{+}\to \infty$, B vanishes (B(∞)=0). When the parameter σ increases, B(0) also increases. With the aid of Equations (19), (24) and (26), we find the generalized Smarr relation
(27)M=2ST−2PV+qΦ+2βB.

## 4. Thermodynamics of Black Holes

To study the local stability of black holes, one can analyze the heat capacity
(28)$$C_{q}=T{\left(\frac{\partial S}{\partial T}\right)}_{q}=\frac{T\partial S/\partial r_{+}}{\partial T/\partial r_{+}}=\frac{2\pi r_{+}T}{\partial T/\partial r_{+}}.$$
Equation (Equation 28) shows that when the Hawking temperature has an extremum, the heat capacity possesses a singularity and the black hole phase transition occurs. With the help of Equation (Equation 24), we depict in Figure 3 the Hawking temperature as a function of the event horizon radius.

For the case σ=1, the analysis of a black hole’s local stability was performed in [21]. The behavior of *T* and $C_{q}$ depends on many parameters. By virtue of Equation (Equation 24), we obtain
$$\frac{\partial T}{\partial r_{+}}=\frac{1}{4\pi }[-\frac{1}{r_{+}^{2}}+\frac{3}{l^{2}}+\frac{3q^{2}}{r_{+}^{4}}F\left(\sigma ,\frac{1}{4};\frac{5}{4};-\frac{q^{2}\beta }{r_{+}^{4}}\right)-\frac{32\sigma q^{4}\beta }{5r_{+}^{8}}F\left(\sigma +1,\frac{5}{4};\frac{9}{4};-\frac{q^{2}\beta }{r_{+}^{4}}\right)$$
(29)$$+\frac{16q^{6}\beta^{2}\sigma \left(4\sigma +1\right)}{9r_{+}^{12}}F\left(\sigma +2,\frac{9}{4};\frac{13}{4};-\frac{q^{2}\beta }{r_{+}^{4}}\right)].$$
Equations (24) and (29) define the heat capacity (28). Making use of Equations (24), (28) and (29), one can study the heat capacity and the black hole phase transition for different parameters β, σ, *q* and *l*.

With the help of Equation (Equation 24), we obtain the black hole equation of state (EoS):(30)$$P=\frac{T}{2r_{+}}-\frac{1}{8\pi r_{+}^{2}}+\frac{q^{2}}{8\pi r_{+}^{4}}\left[F\left(\sigma ,\frac{1}{4};\frac{5}{4};-\frac{q^{2}\beta }{r_{+}^{4}}\right)-\frac{4q^{2}\beta \sigma }{5r_{+}^{4}}F\left(\sigma +1,\frac{5}{4};\frac{9}{4};-\frac{q^{2}\beta }{r_{+}^{4}}\right)\right].$$
The specific volume is given by $v=2l_{P}r_{+}$ ($l_{P}=\sqrt{G}=1$) [11]. Equation (Equation 30) is similar to the EoS of the Van der Waals liquid. Placing $v=2r_{+}$ into expression (30), we obtain
$$P=\frac{T}{v}-\frac{1}{2\pi v^{2}}+\frac{2q^{2}}{\pi v^{4}}[F\left(\sigma ,\frac{1}{4};\frac{5}{4};-\frac{16q^{2}\beta }{v^{4}}\right)$$
(31)$$-\frac{64q^{2}\beta \sigma }{5v^{4}}F\left(\sigma +1,\frac{5}{4};\frac{9}{4};-\frac{16q^{2}\beta }{v^{4}}\right)].$$

The plot of *P* vs. *v* is given in Figure 4.

The critical points (inflection points) are defined by the equations ∂P/∂v=0, $\partial^{2}P/\partial v^{2}=0$, which are complex, so we will not present them here. The analytical solutions for critical points do not exist. The P−v diagrams at the critical values are similar to Van der Waals liquid diagrams having inflection points.

Because *M* is treated as a chemical enthalpy, the Gibbs free energy reads
(32)G=M−TS.
Making use of Equations (19), (20), (24) and (32), we obtain
$$G=\frac{r_{+}}{4}-\frac{2\pi r_{+}^{3}P}{3}+\frac{3q^{2}}{4r_{+}}F\left(\sigma ,\frac{1}{4};\frac{5}{4};-\frac{q^{2}\beta }{r_{+}^{4}}\right)$$
(33)$$-\frac{q^{4}\beta \sigma }{5r_{+}^{5}}F\left(\sigma +1,\frac{5}{4};\frac{9}{4};-\frac{q^{2}\beta }{r_{+}^{4}}\right).$$
The plot of *G* versus *T* is given in Figure 5 for β=0.1, q=1, σ=0.75.

The critical points and phase transitions of black holes for σ=1 were studied in [21]. One can investigate black hole phase transitions in our model for an arbitrary σ with the help of the Gibbs free energy (33). It should be noted that the analytical expressions obtained can be applied for black holes of any sizes. In Figure 1, Figure 2, Figure 3, Figure 4 and Figure 5, we consider examples only for tiny black holes (quantum black holes).

## 5. Summary

We have obtained magnetic black hole solutions in Einstein–AdS gravity coupled to NED with two parameters, which we propose here. The metric and mass functions and their asymptotics with corrections to the Reissner–Nordström solution, when the cosmological constant is zero, have been found. The total black hole mass includes the Schwarzschild mass and the magnetic mass, which is finite. We have plotted the metric function showing that black holes may have one or two horizons. When parameter σ increases, the event horizon radius $r_{+}$ decreases. Figure 2, Figure 3 and Figure 4 show how other physical variables depend on σ. The black hole thermodynamics in an extended phase space was studied. We formulated the first law of black hole thermodynamics where the pressure is connected with the negative cosmological constant (AdS spacetime) conjugated to the Newtonian geometric volume of the black hole. The thermodynamic potential conjugated to the magnetic charge and the thermodynamic quantity conjugated to coupling β (so called vacuum polarization) were computed and plotted. It was proven that the generalized Smarr relation holds for any parameter σ. We calculated the Hawking temperature, the heat capacity and the Gibbs free energy. Analyses of the first-order and second-order phase transitions were performed for some parameters. The Gibbs free energy showed the critical ‘swallowtail’ behavior, which is similar to the Van der Waals liquid–gas behavior. Figure 5 shows a first-order phase transition with Gibbs free energy that is continuous but not differentiable, but, for a second-order transition, the Gibbs free energy and its first derivatives are continuous. The same feature was first discovered for another model in [11]. It was shown within the NED proposed that the electric fields of charged objects at the origin and the electrostatic self-energy are finite. It should be noted that the first law of electric black hole thermodynamics in Einstein–Born–Infeld theory and other problems were originally studied in Ref. [11].

## Figures and Tables

**Figure 1 entropy-26-00261-f001:**
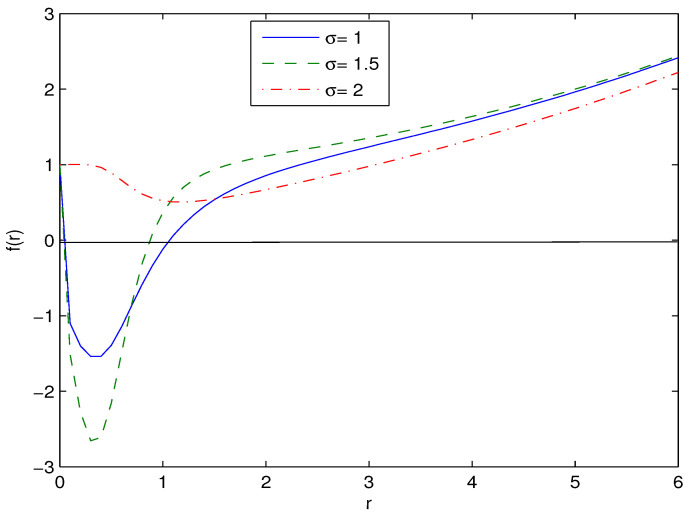
The function f(r) at $m_{0}=0$, G=1, q=1, β=0.1, l=5. Figure 1 shows that black holes may have one or two horizons. When σ increases, the event horizon radius $r_{+}$ decreases.

**Figure 2 entropy-26-00261-f002:**
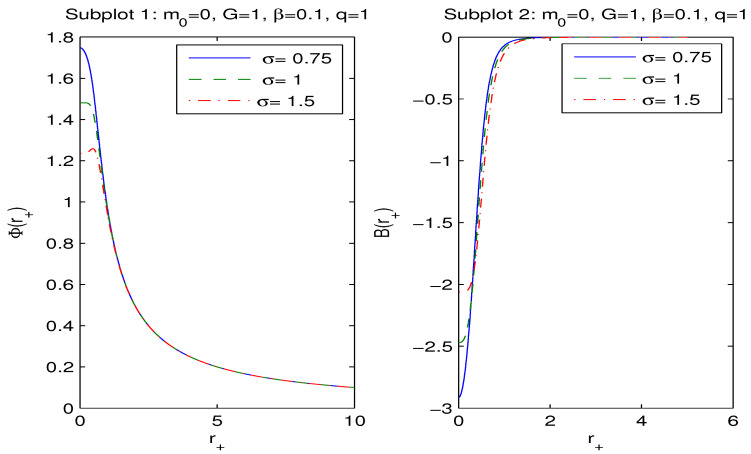
The functions Φ and B vs. $r_{+}$ at q=1, β=0.1. The solid curve in subplot 1 is for σ=0.75, the dashed curve is for σ=1, and the dashed-dotted curve is for σ=1.5. It follows that the magnetic potential Φ is finite at $r_{+}=0$ and becomes zero as $r_{+}\to \infty$. The function B, in subplot 2, vanishes as $r_{+}\to \infty$ and is finite at $r_{+}=0$.

**Figure 3 entropy-26-00261-f003:**
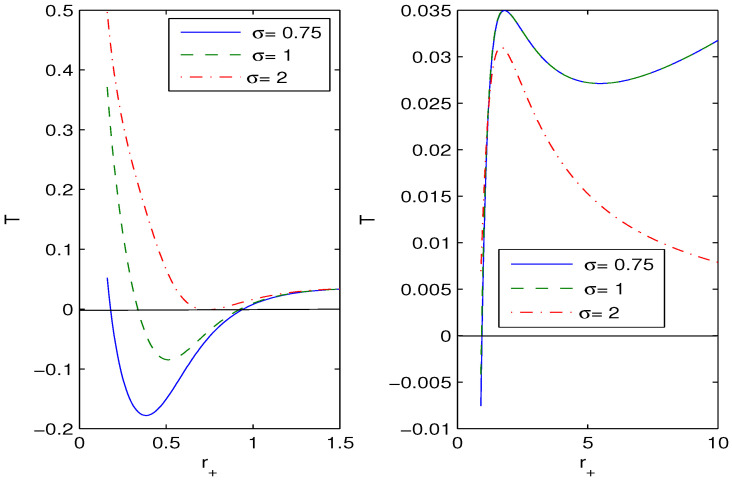
The functions *T* vs. $r_{+}$ at q=1, β=0.1, l=10. The solid curve in the left panel is for σ=0.75, the dashed curve is for σ=1, and the dashed-dotted curve is for σ=2. In some intervals of $r_{+}$, the Hawking temperature is negative and, therefore, black holes do not exist at these parameters. There are extrema of the Hawking temperature *T* where the black hole phase transitions occur.

**Figure 4 entropy-26-00261-f004:**
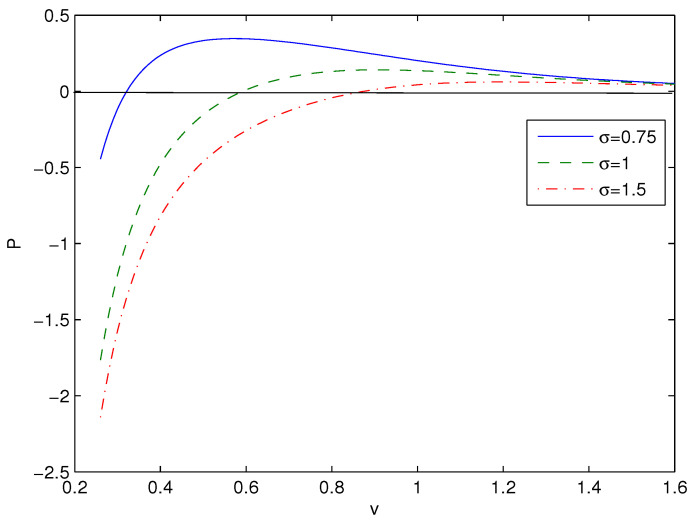
The functions *P* vs. *v* at q=1, β=0.1, T=0.05. The solid line is for σ=0.75, the dashed curve is for σ=1, and the dashed-dotted curve is for σ=1.5.

**Figure 5 entropy-26-00261-f005:**
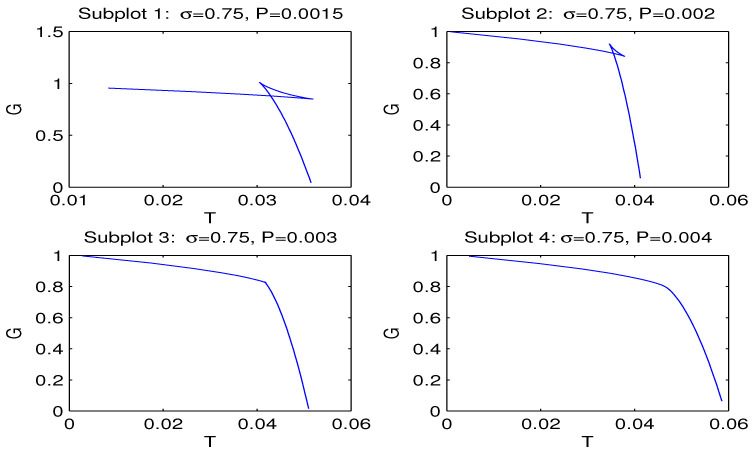
The functions *G* vs. *T* at q=1, β=0.1, σ=0.75 for *P* = 0.0015, *P* = 0.002, *P* = 0.003 and *P* = 0.004. Subplots 1 and 2 show the critical ’swallowtail’ behavior with first-order phase transitions between small and large black holes. Subplot 3 corresponds to the case of critical points where a second-order phase transition occurs ($P_{c}\approx 0.003$). Subplot 4 shows the non-critical behavior of the Gibbs free energy.

## Data Availability

Data sharing is not applicable to this article.

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
