# Peer review of "Magnetic Black Hole Thermodynamics in an Extended Phase Space with Nonlinear Electrodynamics"

_entropy, 2024, doi:10.3390/e26030261_

Round 1

Reviewer 1 Report

Comments and Suggestions for Authors

Comments on the Quality of English Language

Author Response

The author examines the first law of a magnetic black hole solution in 
the context of NonLinear Electrodynamics (NED). This kind of problem was 
originally addressed in reference [11], but here the author considers a different Lagrangian (2) for NED.  Several comments can be made. First, the choice (2) appears to be  motivated by what was done by the author in references
[14-17]. However there is not a clear physical explanation as to why (2) 
should be of any particular interest. There needs to be a proper 
discussion of this point.

A) We added in Introduction: "The NED model, with coupling $\beta$ and 
dimensionless parameter $\sigma$ proposed here, is  of interest because 
it includes a model [21] for  $\sigma=1$. This united approach allows us to find similarities and differences for different models with different $\sigma$." For example, Fig. 1 shows that when $\sigma$ increases the event 
horizon radius decreases. Also, figures 2-4 show how other physical variables depend on \sigma. We added in Summery: "Figures 2-4  show how other physical variables depend on \sigma, It should be noted that the first law of an electric black hole thermodynamics in the Einstein--Born--Infeld theory and other problems were originally studied in Ref. [11]." The Summary was extended.

Second, nothing has been done to compare to ref [11] or to other more 
recent work on the thermodynamics of NED black holes (a quick perusal of 
INSPIRE indicates that at least 25 papers have appeared on this subject 
in the last 5 years). What particular thermodynamic features of this 
theory stand out relative to other work that has been done in this area?

A) In this paper we consider more general model with two parameters
$\beta$ and $\sigma$. It was shown that for any parameter $\sigma$ there 
are not singularities in the electric field in the center and self-energy
is finite. References were added. 

In addition to this are the following issues:
1. Correct the inconsistency in the notation for Newton’s constant: G in 
line 50 and GN in Eq. (7).

1A) The typo was corrected.

2. Clarify the definition and significance of the parameter γ, as it does
 not appear in Eq.(2).

2A) The typo was corrected. The \gamma was replaced by \sigma.

3. Provide additional explanation on deriving ρM between Eqs. (9) and 
(10) and elaborate on the concept of the electromagnetic 4-vector.

3A) We added Appendix C., where the explanation and deriving ρ_M were 
provided.

4. Explain the rationale as to why the symbol M represents the ADM mass 
in Eq. (17).

4A) In (17) $M = m_0+m_M$ is just a notation. Eq. (17) shows that 
$m_0+m_M$ is the ADM mass.

5. Address the absence of the one-horizon case in Fig. 1 and clarify the 
relationship between σ and the event horizon radius r+.

5A) We changed values of \sigma in Fig. 1 showing that for some parameters
there is one or two event horizons. Also, we added after Fig. 1: "It 
should be noted that when we set 
$G=c=\hbar=1$ as in Fig. 1, we come to Planckian units \cite{Mukhanov}. 
Then in this case if one have, for example, dimensionless event horizon 
radius $r_+=1$ (as in Fig. 1) , in usual units 
$r_+=l_{Pl}=(G\hbar/c^3)^{1/2}=1.616\times 10^{-33}$ cm, where 
$l_{Pl}$ is Planck's length. When the dimensionless mass is $m=2$, for 
example, in usual units $m=2\times m_{Pl}= 2\times 2.177\times 10^{-5}$ 
g, where $m_{Pl}$ is Planck's  mass. Because in Fig. 1 the event horizon 
radius is small we have here the example of tiny black holes (primordial 
black holes). Such black holes could be just created after Big Bang." 

6. Correct the pressure in line (95) to P = −Λ/8πl^2.

6A) The pressure is P = −Λ/8π, Λ=-3/l^2.

7. Ensure consistency in the dimensional notation for the coupling β : 
[L] = 4 in line 54 and [β] = L^2 in line 99.

7A) We added after Eq. (18): It is worth noting that the charge $q$ is 
dimensionless and $\beta$ has the dimension $[\beta]=L^4$ when $G\neq 1$, 
but if one uses units with $G=1$, we have $[q]=L$ and $[\beta]=L^2$.

8. Explain the presence of J in line 98 and Eq. 18.

8A) It was written after (18): "J is the black hole angular momentum.
In the following we consider non-rotating black holes and, therefore, 
J = 0."

9. In Eq. 19, P has been introduced in line 95.

9A) We removed P in Eq. 19.

10. Reorganize the appendices to follow the summary section for improved 
coherence and relevance to the main topic.

10A) The appendices were reorganized.

11. Provide a clear description of the study’s physical implications and 
significance in both the introduction and summary sections.
Overall, the paper is not particularly well written (for example the 
appendix appears before the final section). It at least needs a major 
revision before it can be considered further.

11A) We extended Introduction, Conclusion and added references. Appendices were put
after the final section. We made a major revision.

Reviewer 2 Report

Comments and Suggestions for Authors

Author Response

1) After Eq. (2) the author talks about a dimensionless parameter \gamma 
that I did not find in the rest of the manuscript.

1A)This typo was corrected: instead \gamma we wrote \sigma.

2)Usual problem with nonlinear electrodynamics: the total mass of the 
black hole, M =m_0 + m_M, is not just given by a free parameter (m_0) but
depends on a parameter of the Lagrangian,...This needs to be explained.

2A) After Eq. (11) we added: "Equation (11) shows that at the Maxwell's 
limit $\beta\rightarrow 0$ the black hole magnetic mass diverges. 
Therefore, a smooth limit to Maxwell's theory is questionable."

3)...So apparently we are in the unit system of $\hbar$ = 1. This should be 
mentioned at the beginning of the manuscript.

3A) At the end of Introduction we added: "We use units with $c=\hbar=1$."

4) ...For that reason, one should rather set 4s - 2 > 1 and hence arrive 
at s > 3/4. This means that two choices of s in Fig. 1 are actually 
singular black holes. This should be changed.

4A) We added after (14): "Equation (14) shows that at $\sigma\geq 1/2$ a 
singularity of the metric function $f(r)$ is absent. In addition, to 
avoid  a conical singularity at $r=0$  we also should set $4\sigma- 2 > 1$ 
($\sigma>3/4$. It worth noting that the magnetic energy density 
$\rho_M$ is finite at $r=0$ only if $\sigma\geq 1$. Therefore, to have 
regular black holes one has to assume that $\sigma\geq 1$." We also 
changed Fig. 1.

5)Still on Fig. 1: what is plotted? The exact f(r) or the asymptotic 
form?

5A) The exact f(r) is plotted.

6) ...In other words, what does setting G = 1 and q = 1 imply? Is it 
realistic?

6A) We added after Fig. 1: "It should be noted that when we set 
$G=c=\hbar=1$ as in Fig. 1, we come to Planckian units [29]. 
Then in this case if one have, for example, dimensionless event horizon 
radius $r_+=1$ (as in Fig. 1) , in usual units 
$r_+=l_{Pl}=(G\hbar/c^3)^{1/2}=1.616\times 10^{-33}$ cm, where 
$l_{Pl}$ is Planck's length. When the dimensionless mass is $m=2$, for 
example, in usual units $m=2\times m_{Pl}= 2\times 2.177\times 10^{-5}$ 
g, where $m_{Pl}$ is Planck's  mass. Because in Fig. 1 the event horizon 
radius is small we have here the example of tiny black holes (primordial 
black holes). Such black holes could be just created after Big Bang." 

7) Before Eq. (18) we find that [q] = length which is not in keeping with
 previously chosen units. This makes everything following questionable. 
Please explain.

7A) We added after Eq. (18): "It is worth noting that the charge $q$ is 
dimensionless and $\beta$ has the dimension $[\beta]=L^4$ when $G\neq 1$,
 but if one uses units with $G=1$, we have $[q]=L$ and $[\beta]=L^2$."

8) After Eq. (30) the author talks about the singular limit ß ? 0 which 
leads to a divergent black hole mass. I do not understand why this is 
meaningful.

8A) I have removed this sentence.

9) It was not demonstrated that the presented metric is indeed a solution
of the Einstein-Maxwell system. Please insert this calculation in an 
appendix, since it is a central claim of the manuscript that this is 
indeed a solution.

9A) In Appendix C we obtain the metric which is a solution of the 
Einstein-Maxwell system.

10) For some reason, the appendix appears in front of the conclusions, 
which is strange.

10A) I have placed the appendices after the conclusions,

Reviewer 3 Report

Comments and Suggestions for Authors

The author considers static spherically symmetric magnetic black hole (BH) solutions of GR with a source in the form of NED with the Lagrangian (2)and a cosmological constant. The solution itself is obtained using a well-known methodology. Then, various properties of such BHs are discussedincluding their thermodynamics. In my opinion, there are a number of questions and remarks to be addressed before the paper could be recommended for publication.

1. The main question is: what new knowledge of particular interest is gained by this work, recalling that quite a large number of similar studies of the GR-NED system are known in the literature (and partly cited here)?

2. After Eq.(14) it is stated that $\sigma\geq 1/2$ is "a necessary condition to have the spacetime regular." It is true, but it is not a sufficient condition: for example, the magnetic energy density is finite at $r=0$ only if $\sigma\geq 1$, hence the geometry can be regular only under the same condition.

3. Some comments are necessary on the meaning of thermodynamic quantities that concern the BH as a single object rather than its local characteristics. For example, the volume in Eq.(19) is equal to a Euclidean inner volume of a sphere of radius $r_+$ whereas in the BH space-time this region is not only curved but even does not belong to the static reference frame. And the "pressure" $P$ has nothing to do with the local pressure involved in the energy-momentum tensor.

4. Using the term "extended phase space," it would be desirable to explain (say, in the introduction), what is the "original," not extended phase space, what extends it and why.

5. After Eq.(33) we read that "One can investigate black holes phase transitions with the help of Gibbs's free energy," but I do not see such a study, it is onlymentione d that such a transition does exist. What are its type, characteristics and consequences?

6. In the two Appendices the author discusses the properties of a possible electric field in the background of the BH. However, nothing is said about a physical picture under consideration. Is it assumed that there is a point electric charge at the center, small enough as not to affect the geometry? Well, but if yes, the electric energy density (B1) diverges at the center, which is hardly compatible with its regularity.

7. Some remarks on the presentation. First, in the abstract we read that BHs are studied in GR-AdS space-time, but then it is said that their asymptotic is compared with the Reissner-Nordstrom metric - which is not asymptotically AdS. Second, the first two sentences in the introduction are grammatically incorrect, so one can only guess what the author wanted to say. What if I prefer, as a reader, not to guess but to know? Third, it is very inconvenient that the figures are far away in the end of the manuscript rather than where they are mentioned in the text.

I conclude that the paper needs deep reworking and subsequent new refereeing.

Comments on the Quality of English Language

In general, moderate editing is needed. However, as mentioned in my report,  the first two sentences in the introduction are grammatically incorrect, so one can only guess what the author wanted to say.

Author Response

The author considers static spherically symmetric magnetic black hole 
(BH) solutions of GR with a source in the form of NED with the Lagrangian
 (2)and a cosmological constant. The solution itself is obtained using a 
well-known methodology. Then, various properties of such BHs are 
discussed including their thermodynamics. In my opinion, there are a 
number of questions and remarks to be addressed before the paper could be 
recommended for publication.

1. The main question is: what new knowledge of particular interest is 
gained by this work, recalling that quite a large number of similar 
studies of the GR-NED system are known in the literature (and partly 
cited here)?

1A) We added in Introduction: "The NED model, with coupling $\beta$ and 
dimensionless parameter $\sigma$ proposed here, is  of interest because 
it includes a model [21] for $\sigma=1$. This united approach allows us to find similarities and differences for different models." 
2. After Eq.(14) it is stated that $\sigma\geq 1/2$ is "a necessary 
condition to have the spacetime regular." It is true, but it is not a 
sufficient condition: for example, the magnetic energy density is finite 
at $r=0$ only if $\sigma\geq 1$, hence the geometry can be regular only 
under the same condition.

2A) It was added after Eq.(14): "Equation (14) shows that at $\sigma\geq 1/2$ a singularity of the metric function $f(r)$ is absent. It worth noting that the magnetic energy density $\rho_M$ is finite at $r=0$ only if $\sigma\geq 1$. 
Therefore, we will assume that $\sigma\geq 1$."   

3. Some comments are necessary on the meaning of thermodynamic quantities
that concern the BH as a single object rather than its local characteris
tics. For example, the volume in Eq.(19) is equal to a Euclidean inner 
volume of a sphere of radius $r_+$ whereas in the BH space-time this 
region is not only curved but even does not belong to the static 
reference frame. And the "pressure" $P$ has nothing to do with the local 
pressure involved in the energy-momentum tensor.

3A) We added in Introduction: " The first law of black hole 
thermodynamics can be formulated within Einstein's gravity if one 
includes the $V dP$ term. This requires to introduce a negative 
cosmological constant $\Lambda$ as a positive pressure 
$P=-\Lambda/(8\pi)$. As a result, we come to AdS space-time. 
It should be noted that thermodynamic pressure $P$ is different from the 
local pressure which is present in the energy-momentum tensor. The 
conjugate variable to $P$ is the thermodynamic volume $V = 4\pi r_+^3$, 
where $r_+$ is the event horizon radius of a black hole."

4. Using the term "extended phase space," it would be desirable to 
explain (say, in the introduction), what is the "original," not extended 
phase space, what extends it and why.

4A) We added in Introduction: The cosmological constant variation was 
included in the first law of black hole thermodynamics in Refs. 
[12-17]. But within general relativity the cosmological constant $\Lambda$ is a fixed external parameter. Also, such variation of $\Lambda$ in the first 
law of black hole thermodynamics means consideration of black hole 
ensembles possessing different asymptotics. This point of view is 
different from standard black hole thermodynamics, where parameters 
are varied but AdS background is fixed. There are some reasons to 
consider the variation of $\Lambda$ in black hole thermodynamics. 
First of all physical constants can arise as vacuum expectation values 
and not fixed apriori, and therefore, may vary. As a result, these 
‘constants’ are not real constants and may be included in the first law 
of black hole thermodynamics [18,19]. The second reason
 is that without varying the cosmological constant the Smarr relation is 
inconsistent with the first law of black hole thermodynamics 
[13]. When $\Lambda$ is inserted in the first law of black hole 
thermodynamics, the mass M of black holes should be treated as enthalpy 
but not than internal energy [13]." Refs. were added.

5. After Eq.(33) we read that "One can investigate black holes phase 
transitions with the help of Gibbs's free energy," but I do not see such 
a study, it is only mentioned that such a transition does exist. What 
are its type, characteristics and consequences?

5A) It was written in Caption of Fig. 5: "Subplots 1 and 2 show the 
critical 'swallowtail' behavior with first-order phase transitions 
between small and large black holes. Subplots 3 corresponds to the case 
of critical point where second-order phase transition occurs. Subplots 4 
shows noncritical behavior of the Gibbs free energy." Thus, we studied 
first-order and second-order phase transitions only for $q =1,\beta= 0,1, 
\sigma=0.75$. 

6. In the two Appendices the author discusses the properties of a 
possible electric field in the background of the BH. However, nothing is
 said about a physical picture under consideration. Is it assumed that 
there is a point electric charge at the center, small enough as not to 
affect the geometry? Well, but if yes, the electric energy density (B1) 
diverges at the center, which is hardly compatible with its regularity.

6A) At the end of Appendix A we added: "Because of nonlinearity of  
electric fields, an electric charge is not a real point-like particle and does not possesses a singularity at the center." 

7. Some remarks on the presentation. First, in the abstract we read that 
BHs are studied in GR-AdS space-time, but then it is said that their 
asymptotic is compared with the Reissner-Nordstrom metric - which is not 
asymptotically AdS. Second, the first two sentences in the introduction 
are grammatically incorrect, so one can only guess what the author 
wanted to say. What if I prefer, as a reader, not to guess but to know? 
Third, it is very inconvenient that the figures are far away in the end 
of the manuscript rather than where they are mentioned in the text.

7A) In the abstract we made a correction: "We obtain the mass and metric 
functions, their asymptotic and corrections to the Reissner-Nordstrom 
metric function when the cosmological constant vanishes." The first two 
sentences in the introduction were corrected. The figures were placed
in the text  where they are mentioned.

I conclude that the paper needs deep reworking and subsequent new 
refereeing. In general, moderate editing is needed. However, as mentioned in my report,  the first two sentences in the introduction are grammatically 
incorrect, so one can only guess what the author wanted to say.

8A) We made moderate editing and added references.

Round 2

Reviewer 1 Report

Comments and Suggestions for Authors

The author has made substantive revisions to the paper, which have considerably improved it.  I think the only revision that is further needed concerns the statement after (18), which indicates that the dimensionality of certain variables depends on the choice of G.  This is misleading.  With the canonical choice of G=c=hbar=1, the dimensionality of M=[L], q = [L], beta=[L^2], etc are fixed.  But this really means that it is GM/c^2 = [L], etc that are the quantities that have dimension -- the dimension doesn't change if we depart from the canonical choice of G=c=hbar=1.  The highlighted statement after (18) needs to be revised to reflect this.

Comments on the Quality of English Language

Basically fine -- only minor typos need correction.

Author Response

The author has made substantive revisions to the paper, which have 
considerably improved it.  I think the only revision that is further 
needed concerns the statement after (18), which indicates that the 
dimensionality of certain variables depends on the choice of G.  This is 
misleading.  With the canonical choice of G=c=hbar=1, the dimensionality 
of M=[L], q = [L], beta=[L^2], etc are fixed.  But this really means that
 it is GM/c^2 = [L], etc that are the quantities that have dimension -- 
the dimension doesn't change if we depart from the canonical choice of 
G=c=hbar=1.  The highlighted statement after (18) needs to be revised to 
reflect this.

A) The highlighted statement after (18) was removed. Before (18) we added:
"In the following we will use Planckian units with $G=c=\hbar=1$."

Reviewer 2 Report

Comments and Suggestions for Authors

I am mostly happy with the author's responses. I would like to point out that the text after Fig. 1 is still misleading. Certainly one can work with Planckian units. But it is not clear at all that such a quantum-sized black hole is described by semiclassical gravity as is the subject of this paper. As it stands, this last point needs to be improved.

It is actually quite simple to improve: one needs to plot over dimensionless distances by identifying a suitable distance scale in the problem, which could be the cosmological constant curvature scale \Lambda = -1/L^2, or it could be the gravitational radius r=2GM, it is up to the author to decide what makes sense. The regularization parameter \beta can then be expressed as a dimensionless quantity times one of those scales.

This takes some time to set up properly, but care should be taken when doing so. Otherwise, we end up with somewhat nonsensical diagrams describing quantum black holes in a semiclassical theory, which makes no sense. For example, the mention of primordial black holes is unwarranted here, since those black holes are thought to exist in the early universe, yes, but certainly their cosmological constant is not negative.

Similar comments apply to the Figures 2-5 in the manuscript.

This needs to be addressed. After that (including a proper treatise of the other referees' comments) I am happy to reconsider this manuscript for publication in Entropy.

Author Response

I am mostly happy with the author's responses. I would like to point out 
that the text after Fig. 1 is still misleading. Certainly one can work 
with Planckian units. But it is not clear at all that such a quantum-
sized black hole is described by semiclassical gravity as is the subject 
of this paper. As it stands, this last point needs to be improved.

A1) After Fig. 1 we added: "It is worth noting that such an example of 
quantum-sized black holes are described here by semiclassical gravity."

It is actually quite simple to improve: one needs to plot over 
dimensionless distances by identifying a suitable distance scale in the 
problem, which could be the cosmological constant curvature scale \Lambda
 = -1/L^2, or it could be the gravitational radius r=2GM, it is up to the
 author to decide what makes sense. The regularization parameter \beta 
can then be expressed as a dimensionless quantity times one of those 
scales. This takes some time to set up properly, but care should be taken when 
doing so. Otherwise, we end up with somewhat nonsensical diagrams 
describing quantum black holes in a semiclassical theory, which makes no 
sense. For example, the mention of primordial black holes is unwarranted 
here, since those black holes are thought to exist in the early universe, 
yes, but certainly their cosmological constant is not negative.

A3) It is not nessesary to introduce positive cosmological constant
in the early universe to describe the inflation (see, for example, "Universe 
inflation and nonlinear electrodynamics,  Eur.Phys.J.C 84 (2024) 2, 205).Similar comments apply to the Figures 2-5 in the manuscript.

A4) We added at the end of section 4: "It should be noted that analytical 
expressions obtained can be applied for black holes of any sizes. In 
Figures 1-5 we have considered examples only for tiny black holes 
(quantum black holes)."

Reviewer 3 Report

Comments and Suggestions for Authors

In the revised version of the paper, the author has addressed most of my previous remarks, however, in my view, some points still must be corrected before the paper is accepted. 

Thus, concerning item 6 in my previous report, I can repeat that the electric field energy density blows up as $r \to 0$ - see Eq. (B2), so even though the total electric field energy is finite, its singularity at the center is still present - and a proper comment is desirable. 

Also, to be more precise, it would be better to mention at the end of the introduction ant at the beginning of Appendix C that the solution is there  obtained with a magnetic field - just to separate it from Appendices A and B.

On page 10 we read something about Eq.(30) - but (30) is only some expression rather than an equation.

After addressing these minor remarks, I think, the paper can be published.

Comments on the Quality of English Language

There are many language errors, for example, "black holes... is investigated"in the abstract, "we proof" on page 2 (line 4 from bottom) and so on,to say nothing on misused articles, so the text needs serious refereeing.  

Author Response

In the revised version of the paper, the author has addressed most of my 
previous remarks, however, in my view, some points still must be 
corrected before the paper is accepted. Thus, concerning item 6 in my previous report, I can repeat that the 
electric field energy density blows up as $r \to 0$ - see Eq. (B2), so 
even though the total electric field energy is finite, its singularity 
at the center is still present - and a proper comment is desirable. 

A1) At the end of Appendix A it was written: "Because of nonlinearity 
of electric fields, an electric charge is not a real point-like object 
and does not possess a singularity at the center."

Also, to be more precise, it would be better to mention at the end of 
the introduction ant at the beginning of Appendix C that the solution is 
there  obtained with a magnetic field - just to separate it from 
Appendices A and B.

A2) It was written in the introduction: "We will study magnetic 
black holes and their thermodynamics in the Einstein-AdS gravity in the 
extended phase space." We added at the beginning of Appendix C "We will 
obtain the solution of Einstein's equation for magnetic black holes."On page 10 we read something about Eq.(30) - but (30) is only some 
expression rather than an equation.

A3) We replaced  "Putting $v=2r_+$ into Eq. (30)..." by  "Putting 
$v=2r_+$ into expression (30)..."